# Effect of Backing Plate Materials in Micro-Friction Stir Butt Welding of Dissimilar AA6061-T6 and AA5052-H32 Aluminum Alloys

**SeongHwan Park, YoungHwan Joo and Myungchang Kang \***

Graduate School of Convergence Science, Pusan National University, Busan 46241, Korea;
S.H.Park.pcthel@naver.com (S.P.); Y.H.Joo.zeronation@nate.com (Y.J.)

\* Correspondence: kangmc@pusan.ac.kr; Tel.: +82-51-510-7395

**Abstract:** Thin sheets of lightweight aluminum alloys, which are increasingly used in automotive, aerospace, and electronics industries to reduce the weight of parts, are difficult to weld. When applying micro-friction stir welding (μ-FSW) to thin plates, the heat input to the base materials is considerably important to counter the heat loss to the jig and/or backing plate. In this study, three different backing-plate materials—cordierite ceramic, titanium alloy, and copper alloy—were used to evaluate the effect of heat loss on weldability in the μ-FSW process. One millimeter thick AA6061-T6 and AA5052-H32 dissimilar aluminum alloy plates were micro-friction stir welded by a butt joint. The tensile test, hardness, and microstructure of the welded joints using a tool rotational speed of 9000 rpm, a welding speed of 300 mm/min, and a tool tilting angle of 0° were evaluated. The heat loss was highly dependent on the thermal conductivity of the backing plate material, resulting in variations in the tensile strength and hardness distribution of the joints prepared using different backing plates. Consequently, the cordierite backing plate exhibited the highest tensile strength of 222.63 MPa and an elongation of 10.37%, corresponding to 86.7% and 58.4%, respectively, of those of the AA5052-H32 base metal.

**Keywords:** micro-friction stir welding; high rotational speed; dissimilar aluminum materials; backing plate; mechanical properties

## 1. Introduction

Recently, aluminum alloy parts have been increasingly used in the electronics industry and for transportation in the fields of automobile, rail, ship, and aerospace manufacturing to achieve weight reduction and the associated energy savings [1–4]. However, in most of these applications, the metal parts are joined by welding, which is challenging when using aluminum alloys, owing to their high coefficient of thermal expansion and high thermal and electrical conductivity [5]. In particular, welding is more difficult for thin plates of aluminum, with a thickness of 1 mm or less. To overcome the challenges of welding thin plates, micro-friction stir welding micro (μ-FSW) was introduced [6].

Micro-friction stir welding (μ-FSW) uses the frictional heat and stirring force generated between a rotating tool and the base material to form a weld. During the μ-FSW process, the μ-FSW tool is rotated and the probe is plunged into the boundary of the adjoining base plates. The tool shoulder should be in intimate contact with the plates during welding. The heat generated by friction between the rotating tool and the plates promotes a local increase in temperature and softens the materials under the tool shoulder. At the same time, the plunged rotating probe moves and mixes the softened materials by intense plastic deformation, welding both in a solid-state weld [7]. This method is considered to be an eco-friendly welding method, as it produces no fumes from spatter or arc flashes [8]. Important

parameters during μ-FSW include the tool rotational speed, welding speed, tool shape, tool tilting angle, and the material of the backing plate (BP). In particular, μ-FSW requires a large amount of energy per unit volume, as it is usually used to weld thin plates (≤1 mm or less) [6]. It is important to optimize the energy input during μ-FSW and ensure a sufficiently high energy per unit volume to achieve a high-quality weld, while minimizing the loss of generated energy. During the μ-FSW process, heat energy is mainly lost to the BP due to the thermal conduction of the BP, which is called heat loss [9].

The Al-Mg-Si series (AA6xxx) and Al-Mg series (AA5xxx) aluminum alloys are used in many fields [10]. The AA6xxx alloys are widely used in various industries due to their excellent workability, corrosion resistance, and low cost. The AA5xxx alloys are used for aircraft fuel and oil lines, fuel tanks, and automobile panels due to their excellent composition. To take advantage of the properties of both alloy types, the welding of dissimilar materials is required to assemble parts. It is difficult to achieve a good weldability with dissimilar materials using μ-FSW due to the different material properties.

Though many studies have been conducted on FSW, few studies have investigated the effect of the BP on the mechanical properties of parts prepared using a high rotational speed during welding. The choice of BP material for μ-FSW of thin sheets with butt joints has not been investigated systematically, although it is a crucial factor determining the heat loss during welding.

In this study, AA6061-T6 and AA5052-H32 aluminum alloy sheets with a thickness of 1 mm were used, as such materials are widely used in electronic packaging, battery cover plates, and automotive parts. To evaluate the effect of the heat loss according to the thermal conductivity of the BP under the same heat input conditions and analyze the corresponding weldability, the rotational speed of the tool and the welding speed were fixed. The BP of the μ-FSW system was applicable, and three materials with different thermal conductivities were used. Consequently, we propose a backing-plate material suitable for the μ-FSW of dissimilar materials.

## 2. Materials and Methods

The relationship between heat loss and weldability in the μ-FSW process was studied with the change of the BP under the same tool rotational speed and welding speed. The mechanism of heat loss to the BP is shown in Figure 1. In the case of a BP with low thermal conductivity, the heat is generated between the rotating tool and base metal, little heat is transferred to the BP, and total heat loss is small. In this case, a large amount of energy per unit volume remains in the weld region, which results in good plastic flow of the molten metal, and then an excellent weld joint can be obtained. Conversely, BPs with high thermal conductivity result in high heat loss, where weak plastic flow may cause defects in the weld joint.

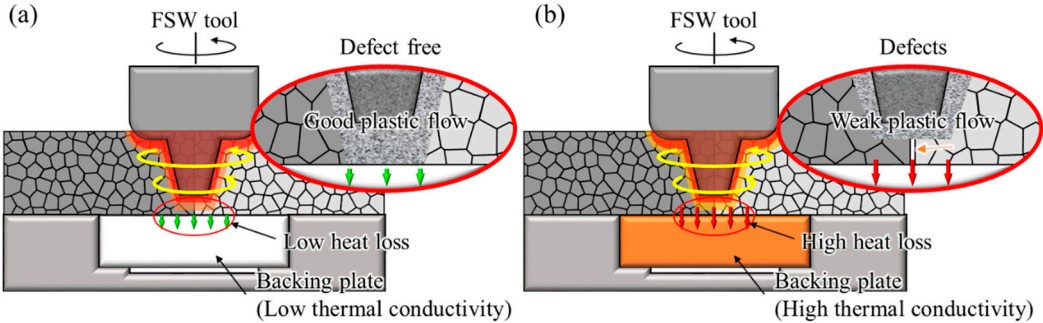

**Figure 1.** Schematic illustration of the mechanism of heat loss to the backing plate using a backing plate with (**a**) a low thermal conductivity and (**b**) a high thermal conductivity.

Three materials with different thermal conductivities were used as BPs under the same heat input conditions to understand the effect of heat loss on weldability. The thermal conductivity and density values of the cordierite ceramic, titanium alloy, and copper alloy materials are shown in Table 1.

**Table 1.** Thermal conductivity and density of backing plate (BP) materials.

| BP Material | Thermal Conductivity (W/(mK)) | Density (g/cm$^3$) |
|---|---|---|
| Cordierite Ceramic | 2.3 | 2.3 |
| Titanium Alloy | 4.48 | 15.6 |
| Copper Alloy | 9.96 | 388.0 |

The materials used in the μ-FSW process were 1 mm thick AA6061-T6 and AA5052-H32 sheets with the same dimensions of 150 mm × 100 mm. The chemical compositions of these alloys are shown in Table 2. The AA6061-T6 sheet was on the advancing side (AS), where more plastic flow and heat occur, while the AA5052-H32 sheet was on the retreating side (RS) [11].

**Table 2.** Chemical composition of the heat-treated base materials (wt%).

| Alloy | Si | Fe | Cu | Mn | Mg | Zn | Cr | Al |
|---|---|---|---|---|---|---|---|---|
| AA5052-H32 | 0.25 | 0.4 | 0.1 | 0.1 | 2.5 | 0.1 | 0.15 | Bal. |
| AA6061-T6 | 0.66 | 0.48 | 0.29 | 0.15 | 1.02 | 0.05 | 0.20 | Bal. |

Figure 2 shows the schematic images of the μ-FSW process used in this study. The tool tilt angle of 0° was used for butt welding along the rolling direction of the base metal to minimize the reduction in the cross-sectional thickness after welding [12]. For the same heat input condition, the welding parameters were kept constant at a tool rotational speed of 9000 rpm and a welding speed of 300 mm/min for all samples. The plunge depth of the tool is 0.05 mm, the dwell time is 0 s, and the plunge speed is 7 mm/min. The temperature during welding was monitored and recorded at the bottom of the base material using a k-type thermocouple. The tool material was H13 tool steel. The probe was tapered from 3 mm at the root diameter to 2 mm at the tip diameter, with a length of 0.65 mm. Welding with a tilting angle of 0° results in a smaller Z-axis load compared to the use of a tilting angle. To increase the Z-axis load, a convex shoulder with a diameter of 6 mm and a convex angle of 3° was used. To maximize the plastic flow and reflow of the softened material, a triple spiral shape was applied to the convex shoulder.

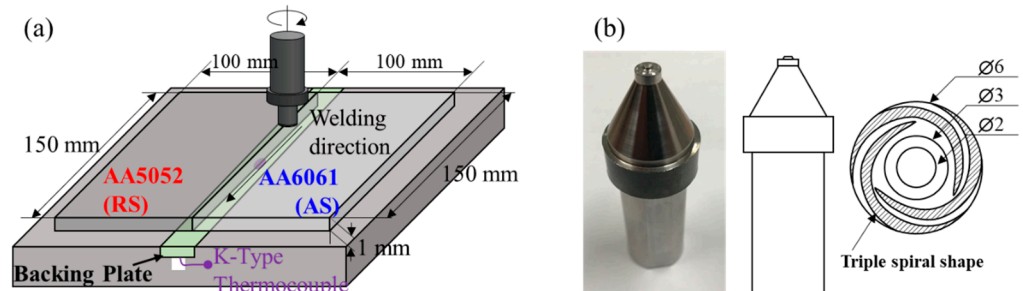

**Figure 2.** Schematic diagrams of (**a**) the overall micro-friction stir welding (μ-FSW) setup and (**b**) the welding tool used in this study.

All samples prepared using dissimilar μ-FSW were cross sectioned perpendicular to the welding direction. For microstructural observation, optical microscopy (OM; KH-8700, HIROX, Tokyo, Japan) and field-emission scanning electron microscopy (FE-SEM; S-4800N, Hitachi, Tokyo, Japan) were used. Specimens for microstructural and mechanical analyses were cut perpendicular to the welding direction using an electrical discharge cutting machine (Sodick, Yokohama, Japan). Tensile specimens were prepared for each joint with reference to ASTM E8 to evaluate the tensile strength of the joint. During the tensile test, elongation was measured using a laser extensometer (Epsilon, Jackson, USA). Hardness measurements of the polished weld cross sections were performed using Vickers indentation

(VMT-X, Matsuzawa, Akita, Japan) at 100 gf and a 10 s dwell time. The polished specimens were etched with Keller's reagent ($HNO_3$ (5 mL) + HCl (0.75 mL) + HF (0.5 mL) + distilled water (43.75 mL)).

## 3. Results

### 3.1. Welding Temperature

The temperature and welding time curves with the three different backing plate materials are shown in Figure 3. The peak temperature measured at the BP depended on its thermal conductivity. The Cu alloy, Ti alloy, and cordierite ceramic BPs showed peak temperatures of 218.5, 277.9, and 292.3 °C, respectively, where the temperature increased with decreasing thermal conductivity of the BP. This is similar to the findings of previous studies [13–15] that also observed a decrease in peak temperature when BPs with high thermal conductivity were used. The cordierite ceramic and Ti alloy BPs showed similar temperature–time curves and peak temperature values. However, when using the Cu alloy BP, the thermal conductivity was very high, so it was measured that the temperature increased due to heat conduction before the tool reached the temperature measuring point. After the peak temperature was reached, the fastest cooling of the welding line was observed for the cordierite ceramic and Ti alloy BPs.

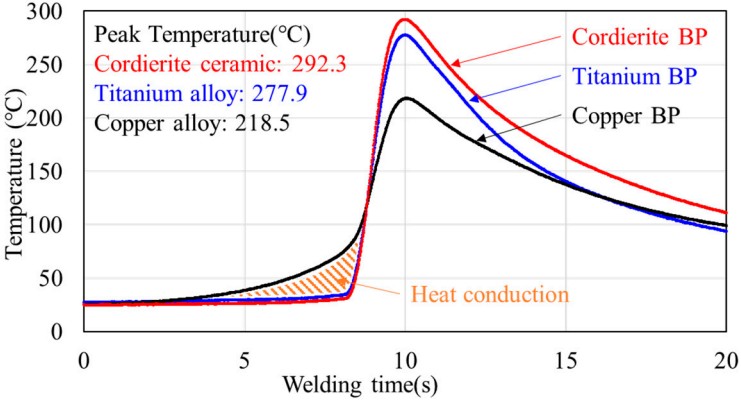

**Figure 3.** Welding temperature at the bottom of the welding line.

### 3.2. Microstructure of the Joint Surface and Cross Section

Figure 4 shows top views of the weld surface, along with the 3D microstructure produced from these images, for µ-FSW with different BP materials. The surface morphology of the welds according to BP was clearly different. Regular surfaces were observed when using the cordierite ceramic and Ti alloy BPs, while the Cu alloy BP resulted in an irregular surface. The higher thermal conductivity of the Cu BP resulted in significant heat loss and irregular material flow during welding.

Figure 5 shows OM and SEM images of the cross-sections of the joints according to the different BP materials. Under all conditions, the section reduction ratio was similar and the butt interface was obvious. When using the cordierite ceramic and Ti alloy BPs, no significant penetration defects, called lack, were observed (see inset SEM images), which was previously attributed to sufficient stirring and welding heat input [16]. However, when the Cu alloy BP was used, penetration defects were manifestly observed. This was attributed to the high heat loss during welding due to the high thermal conductivity of the BP, which resulted in a low welding temperature and reduced stirring ability.

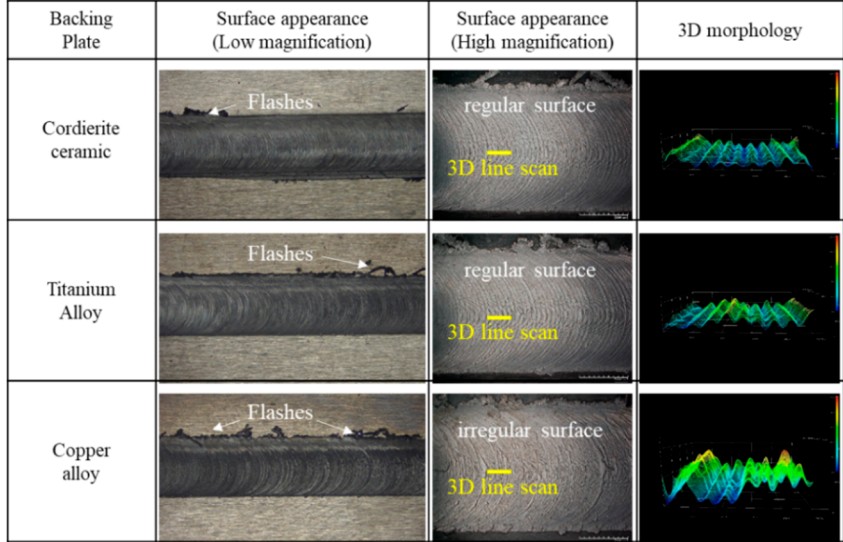

**Figure 4.** Surface morphology of the welds produced with different backing plate (BP) materials.

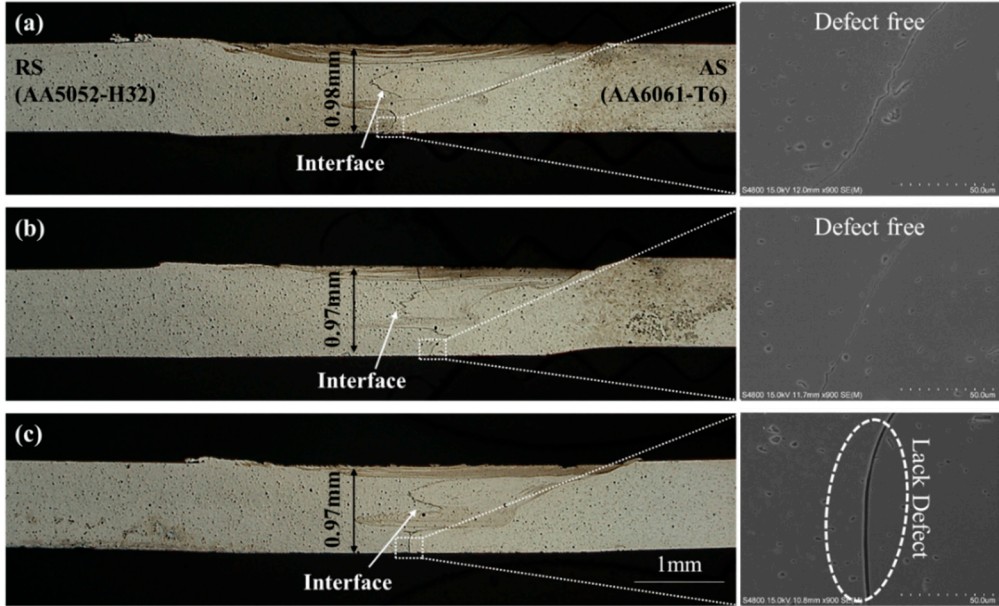

**Figure 5.** Optical microscopy (OM) images of the cross sections of the welding joints prepared using (**a**) cordierite, (**b**) Ti, and (**c**) Cu BP materials, where the insets are scanning electron microscopy (SEM) images of the bottom region, highlighting penetration defects.

*3.3. Mechanical Properties*

The Vickers hardness values measured on the cross sections of all weld specimens are shown in Figure 6a as the distance from the center of the cross section, where the inserted image shows the measurement points. Figure 6b shows hardness mapping results for the entire cross-sectional area. The micro-friction stir welding with the cordierite ceramic and Ti alloy BP materials showed hardness graphs with a "W" shape, where the lowest hardness was observed in the heat-affected zone (HAZ), as observed previously [1,2,13–15].

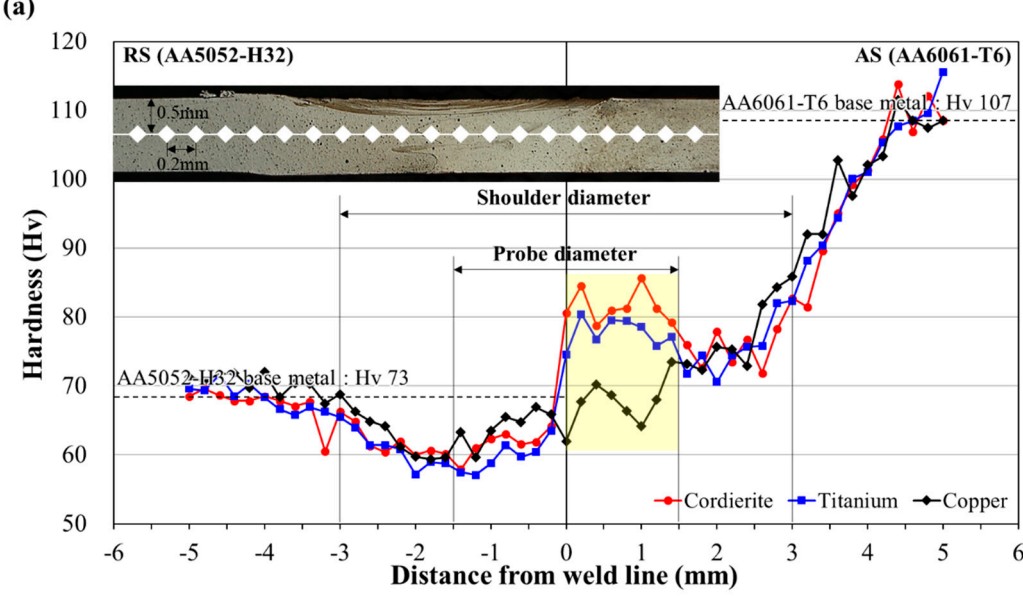

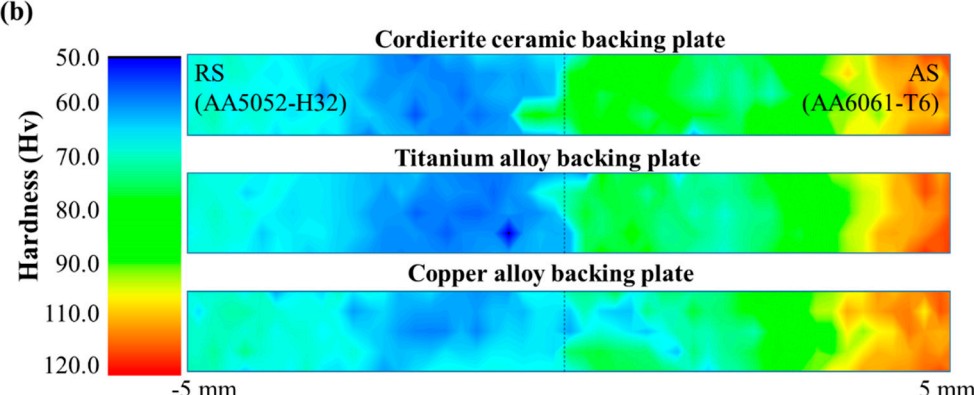

**Figure 6.** (**a**) Hardness profile measured at the center of the cross section and (**b**) microhardness maps of the joints obtained using different BP materials.

Larger temperature deviations are generally observed at the AS of the weld, which has been attributed to the thermal cycle and plastic deformation of the Al–Mg–Si alloy [17]. The joints produced using cordierite ceramic and Ti alloy BP materials, which resulted in the highest welding temperatures, showed higher hardness values, as the precipitate distribution in the stir zone (SZ) was close to that in the base metal (BM) during μ-FSW [15].

### 3.4. Fractography Analysis

Figure 7 shows OM and SEM images of the fractured samples after tensile testing. Figure 7a,b show the fracture location of cordierite ceramic and Ti alloy BP samples, respectively. Both the cordierite and Ti BP samples fractured at the HAZ of the joints, where the fracture was located at the RS near the joint edge. This indicates that the butt interface was well stirred and a high-quality weld was formed. Figure 6c shows the fracture position of the weld produced with the Cu alloy BP. As shown in Figure 4c, a lack of penetration was observed on the bottom surface of the specimen using a Cu alloy BP. Figure 6f shows the cross-section image of the Cu alloy BP sample after tensile testing, which shows that the fracture occurred due to the lack of penetration defects.

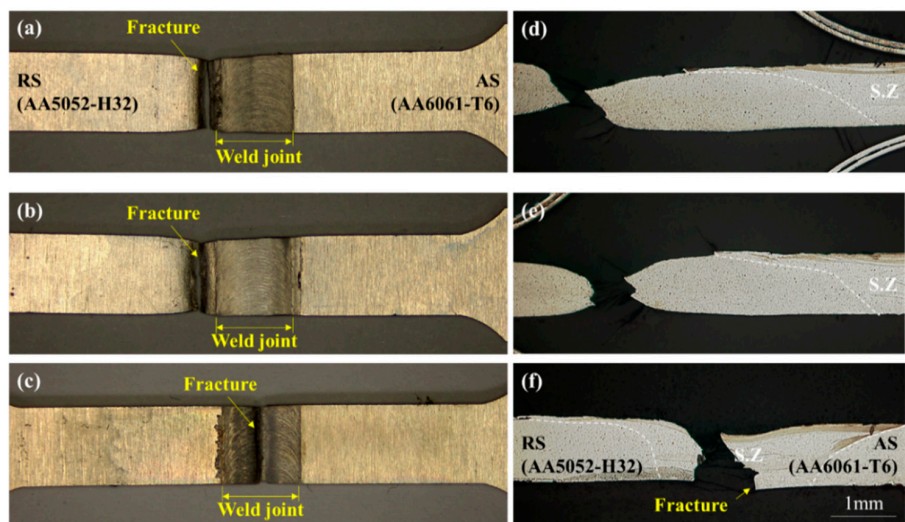

**Figure 7.** (**a**–**c**) OM images and (**d**–**f**) cross-sectional SEM images of the fracture positions of the joints produced using (**a**,**d**) cordierite, (**b**,**e**) Ti, and (**c**,**f**) Cu BP materials.

Figure 8 shows representative stress–strain curves for the three experiments on joints obtained using the different BPs. The tensile strength and elongation of all specimens are listed in Table 3. The mechanical properties of all joints were compared with to those of the base metal (AA5052-H32). The best joint, with respect to the mechanical properties, was obtained using the cordierite ceramic BP, which showed the highest tensile strength of 222.63 MPa and the longest elongation of 10.37%, which were 86.7% and 58.42% of the base metal values, respectively. The low heat input when using the Cu alloy BP led to insufficient plastic flow, which resulted in a joint with poor mechanical properties.

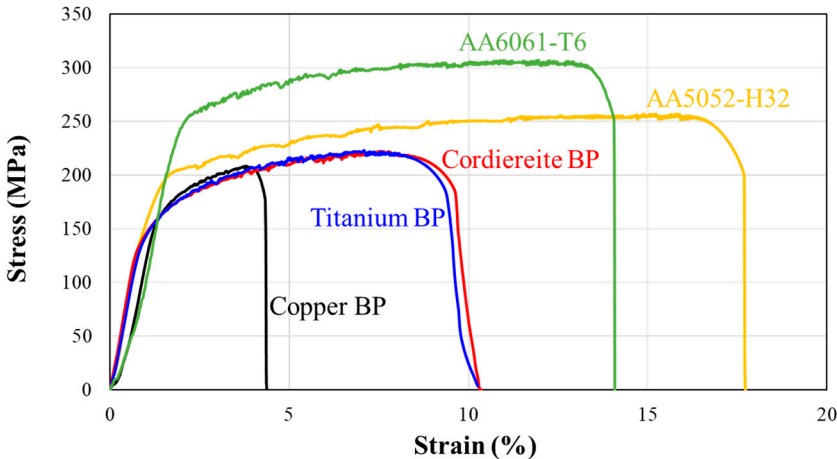

**Figure 8.** Stress–strain curves of the joints obtained using different BP materials.

**Table 3.** Mechanical properties of the joints obtained using different BP materials.

| BP Material | Tensile Strength (MPa) | Elongation (%) |
|---|---|---|
| Base Metal (AA6061-T6) | 306.99 | 14.08 |
| Base Metal (AA5052-H32) | 256.82 | 17.75 |
| Cordierite Ceramic | 222.63 | 10.37 |
| Titanium Alloy | 220.23 | 10.25 |
| Copper Alloy | 208.70 | 4.37 |

The fracture surface morphologies of the base metal and typical joints are shown in Figure 9. As observed in Figure 7, when using the cordierite ceramic and Ti alloy BP materials, ductile fractures

occurred, similar to the base metal. Many large and deep dimples with tearing edges associated with micro-pores were observed in the fracture surface of the base metal, indicating typical ductile fractures. In the case of the joints welded using cordierite and titanium BPs, smaller dimples were observed than those in the base metal, indicating a softening of the joint in the HAZ, as shown in the insets of Figure 9b,c. Figure 9d shows the fracture surface of the sample welded using the Cu alloy BP. There are no obvious dimples in the fracture surface, indicating that no effective bonding occurred in the SZ.

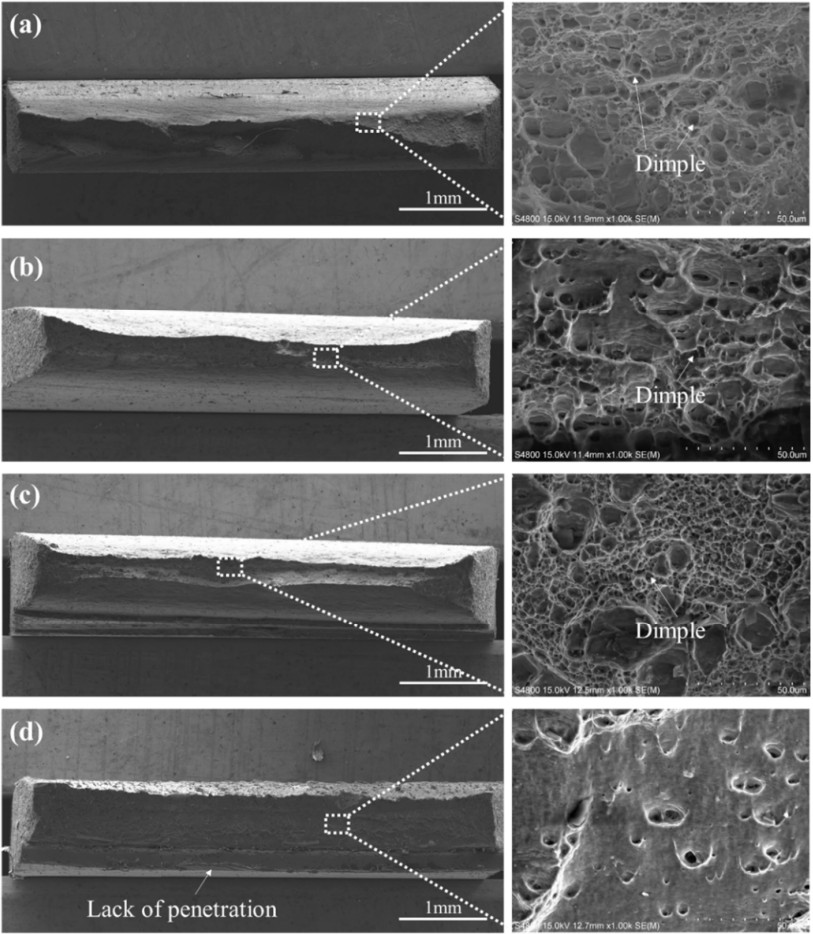

**Figure 9.** Fracture morphologies of the (**a**) AA5052-H32 base metal and μ-FSW joints obtained using (**b**) cordierite, (**c**) Ti, and (**d**) Cu BPs.

## 4. Conclusions

In this work, joints between AA6061-T6 and AA5052-H32 thin plates were successfully produced using dissimilar μ-FSW and the effect of the backing plate material was investigated. The use of a backing plate with lower thermal conductivity, like cordierite ceramic and Ti alloy, resulted in lower heat loss during the welding process. Although the thermal conductivity of the Ti alloy was almost double that of the cordierite ceramic, the peak welding temperature and mechanical properties of the weld were similar. The use of the Cu alloy backing plate with a high thermal conductivity and heat loss resulted in poor stirring at the bottom surface of the welding interface between the materials, resulting in defects and poor mechanical properties. A maximum tensile strength of 222.63 MPa and an elongation of 10.37% were achieved when using the cordierite ceramic backing plate. The fracture morphologies showed typical ductile fractures for the joints prepared using the cordierite and Ti alloy backing plates, similar to the base metal. The results of this study confirm that the choice of the backing plate material is crucial for ensuring the high-quality welding of thin aluminum alloy sheets using μ-FSW.

**Author Contributions:** Conceptualization, writing—original draft preparation, visualization, S.P. and Y.J.; funding acquisition, writing—review and editing, M.K. All authors have read and agreed to the published version of the manuscript.

**Funding:** This research was funded by a National Research Foundation of Korea (NRF) grant funded by the Korean government (MOE) (No. NRF-2018R1D1A1B07051302).

**Conflicts of Interest:** The authors declare no conflict of interest.

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
