# Peer review of "Effect of Backing Plate Materials in Micro-Friction Stir Butt Welding of Dissimilar AA6061-T6 and AA5052-H32 Aluminum Alloys"

_metals, doi:10.3390/met10070933_

Round 1

Reviewer 1 Report

This is a well-written manuscript that presents interesting results of research focusing on the effect of the backing plate material on macrostructure and mechanical properties μ-fristion stir welds. Results of the performed examination were properly showed and rightly discussed.

I recommend the manuscript to publish after one revision: The Authors should be add the sections: 3.4 Fractography analisys and 4 Conclusions.

Author Response

Reviewer #1:

[General review] I recommend the manuscript to publish after one revision:

[Our response] We greatly appreciate the reviewer’s summary and positive evaluation of our manuscript.

[Comment 1] I recommend the manuscript to publish after one revision: The Authors should be add the sections: 3.4 Fractography analisys and 4 Conclusions.

[Our response 1] Accordingly, in revised manuscript text has been modify.

[Modification of the manuscript]

3.4 Fractography analisys

Figure 7 shows OM and SEM images of the fractured samples after tensile testing. Figure 7(a) and Figure 7(b) show the fracture location of cordierite and Ti BP samples respectively.

  1. Conclusion

In this work, joints between AA6061-T6 and AA5052-H32 thin plates were successfully produced using dissimilar μ-FSWs and the effect of the backing plate material was investigated.

[Additional Our response] We got our manuscript corrected by a native speaker. We attached the certificate of English editing as follows.

Finally, we are grateful to the referees for the comments that helped us improve the quality of our manuscript. The related changes in the manuscript were marked with Blue color. We are reply to the referee’s comments on a point by point basis.

We hope that these revisions are satisfactory and that the revised version will be acceptable for publication in Metals.

Thank you very much for your work concerning our paper.

Reviewer 2 Report

It is an interesting paper. It is very legibly written text that is read with pleasure. The topic is appropriate for the journal and the subject is of current interest. Thus, I recommend that manuscript to publish in Metals. However, one improvement should be introduced:

The last paragraph (verses 204 - 216) contain information such as discussion of results and conclusions. There is no header about conclusions only.

After completing the above-mentioned comment, I strongly recommend the paper for publication.

Author Response

Reviewer #2:

[General review] I recommend that manuscript to publish in Metals. However, one improvement should be introduced:

[Our response] We greatly appreciate the reviewer’s summary and positive evaluation of our manuscript.

[Comment 1] The last paragraph (verses 204 - 216) contain information such as discussion of results and conclusions. There is no header about conclusions only.

[Our response 1] Our revised manuscript text has been modify.

[Modification of the manuscript]

  1. Conclusion

In this work, joints between AA6061-T6 and AA5052-H32 thin plates were successfully produced using dissimilar μ-FSWs and the effect of the backing plate material was investigated.

[Additional Our response] We got our manuscript corrected by a native speaker. We attached the certificate of English editing as follows.

Finally, we are grateful to the referees for the comments that helped us improve the quality of our manuscript. The related changes in the manuscript were marked with Blue color. We are reply to the referee’s comments on a point by point basis.

We hope that these revisions are satisfactory and that the revised version will be acceptable for publication in Metals.

Thank you very much for your work concerning our paper.

Reviewer 3 Report

The present paper aims at studying the influence of backing-plate materials in micro FSW of dissimilar AA6061-T6 and AA5052-H32 aluminum alloys.

The paper is interesting but it is not suitable for publication in the present form: additional results have to be shown and many aspects have to be improved.

  • The introduction section is weak and needs to be improved: for example, the reviewer suggests inserting the description of the micro FSW process;
  • The experimental methods in section 2 have to be described through sub-paragraphs;
  • Provide more details about micro FSW operations, such as tool plunging, plunging speed and dwelling time;
  • Was the micro FSW process performed using force control? It would be very interesting to show the trend of the vertical force developed during welding;
  • How was the strain measured during tensile tests? Did you use an extensometer?
  • How many tests have been performed to ensure repeatability of the results? In this regard, enter in Table 3 the deviations of tensile strength and strain;
  • Why did the authors choose the rotational speed and welding speed values of 9000 rpm and 300 mm/min, respectively? Did the authors evaluate the influence of such process parameters on micro- and macro-mechanical properties? It would be very interesting to have information on such results;
  • Add in Figure 8 the stress-strain curve of the base material AA6061-T6;
  • What is the average grain size in the nugget, thermo-mechanically affected zone and heat-affected zone of the joints obtained using the different backing-plate materials?
  • Have the authors studied the effect of the position of the two base materials with respect to the direction of tool rotation? It would be very interesting to show the results obtained by joints welded using AA5052 in the advancing side and AA6061 in the retreating side;
  • The discussion of the results has to be more supported by literature. The reviewer suggests justifying the results by comparing them with those obtained by other authors. Furthermore, the reviewer suggests to read the following papers, useful to support authors' statements:
  • Forcelelse et al. Influence of process parameters on the vertical forces generated during friction stir welding of AA6082-T6 and on the mechanical properties of the joints, Metals (2017), Volume 7.
  • Shunmugasundaram et al. Parametric optimization on tensile strength of friction stir butt joints of dissimilar AA6061 and AA5052 aluminium alloys by Taguchi technique, Materials Today: Proceedings (2020), Vol. 27, 1258-1262
  • Forcellese et al. Similar and dissimilar FSWed joints in lightweight alloys: Heating distribution assessment and IR thermography monitoring for on-line quality control. Key Engineering Materials (2013), Vol. 554-557, 1055-1064

For the aforementioned reasons, the reviewer suggests a major revision.

Author Response

Reviewer #3:

[General review] The paper is interesting but it is not suitable for publication in the present form: additional results have to be shown and many aspects have to be improved.

[Our response] We greatly appreciate the reviewer’s summary and evaluation of our manuscript.

[Comment 1] The introduction section is weak and needs to be improved: for example, the reviewer suggests inserting the description of the micro FSW process;

[Our response 1] Accordingly, in revised manuscript text has been modify.

[Modification of the manuscript]

Micro-Friction stir welding (μ-FSW) uses the frictional heat and stirring force generated between a rotating tool and the base material to form a weld. During the μ-FSW process, the μ-FSW tool is rotated and the probe is plunged into the boundary of the adjoining base plates. Penetration depth of the tool shoulder, which should be in intimate contact with the plates during welding. The heat generated by friction between the rotating tool and the plates promotes a local increase in temperature and softens the materials under the tool shoulder. At the same time, the plunged rotating probe moves and mixes the softened materials, by intense plastic deformation, welding both in a solid state weld. This method is considered to be an eco-friendly welding method, as it produces no fumes from spatter or arc flashes.

[Comment 2] Was the micro FSW process performed using force control? It would be very interesting to show the trend of the vertical force developed during welding;

[Our response 2] We appreciate the reviewer’s helpful comment. We performed the micro-FSW process through position control of Z-axis.

[Comment 3] How was the strain measured during tensile tests? Did you use an extensometer?

[Our response 3] We appreciate the reviewer’s helpful comment. In response to the reviewer’s comment, we revised our manuscript as follows.

[Modification of the manuscript]

During the tensile test, elongation was measured using a laser extensometer.

[Comment 4] How many tests have been performed to ensure repeatability of the results? In this regard, enter in Table 3 the deviations of tensile strength and strain;

[Our response 4] We appreciate the reviewer’s helpful comment. Representative results were expressed through three times. In response to the reviewer’s comment, we revised our manuscript as follows.

[Modification of the manuscript]

Figure 8 shows a representative stress-strain curves for three experiments of joints obtained using the different BPs. The tensile strength and elongation of all specimens are listed in Table 3.

[Comment 5] Why did the authors choose the rotational speed and welding speed values of 9000 rpm and 300 mm/min, respectively? Did the authors evaluate the influence of such process parameters on micro- and macro-mechanical properties? It would be very interesting to have information on such results;

[Our response 5] We appreciate the reviewer’s helpful comment. Prior experiments were performed under tool rotational speed of 6000 to 9000 rpm and welding speed of 150 to 1200 mm/min. A defect occurred due to excessive heat input at a welding speed of 150 mm/min, and a defect was observed due to insufficient heat input at a high welding speed. In order to evaluate the effect of the backing plate at the same tool rotation speed and welding speed, conditions for tool rotation speed 9000 rpm and welding speed 300 mm/min were selected.

[Comment 6] Add in Figure 8 the stress-strain curve of the base material AA6061-T6;

[Our response 6] We appreciate the reviewer’s helpful comment. We revised our manuscript as follows.

[Modification of the manuscript]

[Comment 7] What is the average grain size in the nugget, thermo-mechanically affected zone and heat-affected zone of the joints obtained using the different backing-plate materials?

[Our response 7] We appreciate the reviewer’s helpful comment. Unfortunately, the average grain size of the nugget zone was not confirmed. We will try to make it a useful paper by applying it to the next research.

[Comment 8] Have the authors studied the effect of the position of the two base materials with respect to the direction of tool rotation? It would be very interesting to show the results obtained by joints welded using AA5052 in the advancing side and AA6061 in the retreating side;

[Our response 8] We appreciate the reviewer’s helpful comment. The effect according to the position of the two base materials was confirmed. In addition, it has been previously reported elsewhere that a stronger material should be located in advancing side.

[Comment 9] The discussion of the results has to be more supported by literature. The reviewer suggests justifying the results by comparing them with those obtained by other authors. Furthermore, the reviewer suggests to read the following papers, useful to support authors' statements:

[Our response 9] We appreciate the reviewer’s helpful comment. The proposed paper has been reviewed and will be very helpful. We will try load control system in the next study. Testing using DOE is still being performed. Non-contact measurement and non-destructive method are under consideration. We will try to make it a useful paper by applying it to the next research.

[Additional Comment] English language and style are fine/minor spell check required.

Our response We got our manuscript corrected by a native speaker. We attached the certificate of English editing as follows.

Finally, we are grateful to the referees for the comments that helped us improve the quality of our manuscript. The related changes in the manuscript were marked with Blue color. We are reply to the referee’s comments on a point by point basis.

We hope that these revisions are satisfactory and that the revised version will be acceptable for publication in Metals.

Thank you very much for your work concerning our paper.

Round 2

Reviewer 3 Report

The authors have adequately addressed the reviewer' comments, even if I suggest to take into account the following remarks:

  • the description of the micro-FSW process was provided in the introduction. Since the thickness of the sheets is less than 1 mm or less, it is useful to mention different solutions available in the literature, such as the FSW performed using a pinless tool. I suggest: Simoncini et al. Micro- and Macro- Mechanical Properties of Pinless Friction Stir Welded Joints in AA5754 Aluminium Thin Sheets.
  • Delete the text "Cordierite ceramic, titanium alloy and copper alloy material thermal conductivities are …" (lines 170 -171) because it is redundant. The same information is contained in table 1.
  • The authors replied that "The effect according to the position of the two base materials was confirmed. In addition, it has been previously reported elsewhere that a stronger material should be located in advancing side." Provide references that justify this statement.

The authors did not respond to the following comments requested during the first revision:

  • The experimental methods in section 2 have to be described through sub-paragraphs;
  • Provide more details about micro FSW operations, such as tool plunging, plunging speed and dwelling time.

For the aforementioned reasons, the reviewer suggests a minor revision.

Author Response

Reviewer #3:

[General review] The authors have adequately addressed the reviewer’ comments, even if I suggest to take into account the following remarks:

[Our response] We greatly appreciate the reviewer’s summary and positive evaluation of our manuscript.

[Comment 1] The description of the micro-FSW process was provided in the introduction. Since the thickness of the sheets is less than 1 mm or less, it is useful to mention different solutions available in the literature, such as the FSW performed using a pinless tool.

[Our response 1] The literature you suggested has been thoroughly reviewed. This is a literature that evaluates mechanical properties according to pin tool and pinless tool. The FSW welding of thin plate using ‘pinless’ tool is being conducted in our lab on the mechanical properties of FSWed joints according to the diameter and shape of the shoulder. I think the literature you suggested will be very helpful.

 Unfortunately, this study requires greater stirring power because FSW of differecnt materials is used. Therefore, pin tool was used, and the usefulness of the pinless tool should be studied further.

[Comment 2] Delete the text “Cordierite ceramic. Titanium alloy and copper alloy materials thermal conductivities are …” (line 170 – 171) because it is redundant. The same information is contained in table 1.

[Our response 2] Our revised manuscript text has been modify.

[Modification of the manuscript]

Three materials with different thermal conductivity were used as BPs under the same heat-input conditions to understand the effect of heat loss on weldability. The thermal conductivity and density values of the cordierite ceramic, titanium alloy, and copper alloy materials are shown in Table 1. Cordierite ceramic, titanium alloy and copper alloy material thermal conductivities are 2.3, 4.48, and 9.96 W/(mK), respectively, with density values of 2.3, 15.6, and 388.0 g/cm3, respectively. (delete)

[Comment 3] The authors replied that "The effect according to the position of the two base materials was confirmed. In addition, it has been previously reported elsewhere that a stronger material should be located in advancing side." Provide references that justify this statement.

Our response 3] AA6061-T6 was positioned in the advancing side by referring to the following reference. Babua. N et al. Microstructural and Mechanical Properties of Solid State Welded Dissimilar Aluminum Alloy Joints. Revised manuscript text based on references.

[Modification of the manuscript]

The materials used in the μ-FSW process were 1-mm-thick AA6061-T6 and AA5052-H32 sheets with the same dimensions of 150 mm × 100 mm. The chemical compositions of these alloys are shown in Table 2. The AA6061-T6 sheet was on the advancing side (AS) where more plastic flow and heat occur, while the AA5052-H32 sheet was on the retreating side (RS) [9].

[Comment 4] The experimental methods in section 2 have to be described through sub-paragraphs; Provide more details about micro FSW operations, such as tool plunging, plunging speed and dwelling time.

Our response 4] Our revised manuscript text has been modify.

[Modification of the manuscript]

For the same heat input condition, the welding parameters were kept constant at tool rotational speed of 9000 rpm and welding speed of 300 mm/min for all samples. The plunge depth of the tool is 0.05mm, the dwell time is 0 s, and the plunge speed is 7 mm/min were used.
